# Molecular Pathogenesis of Gene Regulation by the *miR-150* Duplex: *miR-150-3p* Regulates *TNS4* in Lung Adenocarcinoma

**DOI:** 10.3390/cancers11050601

**Published:** 2019-04-30

**Authors:** Shunsuke Misono, Naohiko Seki, Keiko Mizuno, Yasutaka Yamada, Akifumi Uchida, Hiroki Sanada, Shogo Moriya, Naoko Kikkawa, Tomohiro Kumamoto, Takayuki Suetsugu, Hiromasa Inoue

**Affiliations:** 1Department of Pulmonary Medicine, Graduate School of Medical and Dental Sciences, Kagoshima University, Kagoshima 890-8520, Japan; k8574402@kadai.jp (S.M.); keim@m.kufm.kagoshima-u.ac.jp (K.M.); akifumiuchida7883@gmail.com (A.U.); k8173956@kadai.jp (H.S.); kuma@m2.kufm.kagoshima-u.ac.jp (T.K.); taka3741@m2.kufm.kagoshima-u.ac.jp (T.S.); inoue-pulm@umin.net (H.I.); 2Department of Functional Genomics, Graduate School of Medicine, Chiba University, Chuo-ku, Chiba 260-8670, Japan; yasutaka1205@olive.plala.or.jp (Y.Y.); naoko-k@hospital.chiba-u.jp (N.K.); 3Department of Biochemistry and Genetics, Graduate School of Medicine, Chiba University, Chuo-ku, Chiba 260-8670, Japan; moriya.shogo@chiba-u.jp

**Keywords:** MicroRNA, *miR-150-5p*, *miR-150-3p*, lung adenocarcinoma, *TNS4*

## Abstract

Based on our miRNA expression signatures, we focused on *miR-150-5p* (the guide strand) and *miR-150-3p* (the passenger strand) to investigate their functional significance in lung adenocarcinoma (LUAD). Downregulation of *miR-150* duplex was confirmed in LUAD clinical specimens. In vitro assays revealed that ectopic expression of *miR-150-5p* and *miR-150-3p* inhibited cancer cell malignancy. We performed genome-wide gene expression analyses and in silico database searches to identify their oncogenic targets in LUAD cells. A total of 41 and 26 genes were identified as *miR-150-5p* and *miR-150-3p* targets, respectively, and they were closely involved in LUAD pathogenesis. Among the targets, we investigated the oncogenic roles of tensin 4 (*TNS4*) because high expression of *TNS4* was strongly related to poorer prognosis of LUAD patients (disease-free survival: *p* = 0.0213 and overall survival: *p* = 0.0003). Expression of *TNS4* was directly regulated by *miR-150-3p* in LUAD cells. Aberrant expression of TNS4 was detected in LUAD clinical specimens and its aberrant expression increased the aggressiveness of LUAD cells. Furthermore, we identified genes downstream from *TNS4* that were associated with critical regulators of genomic stability. Our approach (discovery of anti-tumor miRNAs and their target RNAs for LUAD) will contribute to the elucidation of molecular networks involved in the malignant transformation of LUAD.

## 1. Introduction

Lung cancer accounts for the largest number of cancer-related deaths in the world and its morbidity increases annually [1]. Lung adenocarcinoma (LUAD) is the most common histological subtype of non-small cell lung cancer (NSCLC) and this type of cancer is the leading cause of cancer-related deaths [2]. The survival rate of lung cancer has increased due to effective treatment strategies, including molecularly-targeted drugs and immune checkpoint inhibitors [3]. However, the efficacy of treatments for patients with distant metastases is limited, thereby resulting in poor prognosis [2]. In LUAD, metastasis occurs even if the primary tumor is small [4,5]. Therefore, the prognosis of advanced LUAD patients is poor, with average 5-year survival rates below 20% [6]. Therefore, additional research to identify novel biomarkers for earlier detection and to develop effective targeted molecular therapies for LUAD is indispensable.

A vast number of studies have revealed that an extremely large number of non-coding RNAs are transcribed from the human genome and actually functional in various cellular processes. Among these non-coding RNAs, microRNAs (miRNAs) are endogenous single-stranded RNA molecules (19–22 nucleotides in length) that function as fine-tuners of RNA expression [7,8,9,10,11]. Importantly, a single miRNA can control a large number of RNA transcripts in normal and disease cells [7,8,9,10,11]. Recent studies showed that aberrant expression of miRNAs closely contributes to the pathogenesis of human diseases, including cancers via disruption of RNA networks [8,9,10,11,12,13]. 

We have been analyzing novel anti-tumor miRNA-mediated oncogenic targets and pathways that contribute to lung tumorigenesis [14,15,16,17,18,19]. Interestingly, our miRNA studies revealed that some passenger strands of miRNAs (e.g., *miR-144-5p* and *miR-145-3p*) derived from miRNA-duplex actually acted as anti-tumor miRNAs in lung squamous cell carcinoma [15,19]. More recently, we demonstrated that two miRNA species derived from *miR-145*-duplex (*miR-145-5p* and *miR-145-3p*) acted as anti-tumor miRNAs in LUAD via targeting several oncogenes [17].

Initially, it was thought that two types of miRNA were derived from double-stranded pre-miRNAs: guide strand miRNAs that control the target genes, and passenger strand miRNAs that lacked function and were degraded [20]. Our studies have altered the conventional understanding of miRNA biogenesis and have shown the importance of exploring passenger strands of miRNAs in cancer cells. 

RNA sequencing-based miRNA expression signatures contribute to a new development of molecular pathogenesis of human cancers. miRNA expression signatures in human cancers revealed that two miRNAs derived from *miR-150*-duplex—*miR-150-5p* (the guide strand) and *miR-150-3p* (the passenger strand)—are frequently downregulated in several types of cancers [21,22,23]. Reports of *miR-150-5p* in cancers are scattered across public databases, but there are few reports of *miR-150-3p*. We have analyzed the anti-tumor activity of *miR-150-3p* in esophageal and head and neck squamous cell carcinomas [22,23]. With regard to LUAD, no such reports are available, and this is an important facet of the functional analysis of *miR-150-3p* and the search for target genes.

The aim of this study was to investigate the anti-tumor roles of both strands of *miR-150*-duplex and to identify their targets with close associations with LUAD tumorigenesis. The Cancer Genome Atlas (TCGA) revealed that low expression of *miR-150-5p* and *miR-150-3p* predicted poor prognosis. Ectopic expression of two miRNAs significantly attenuated the malignant phenotypes of cancer cells. Moreover, we identified several oncogenic targets by *miR-150-3p* regulation in LUAD cells. 

## 2. Results

### 2.1. miR-150-5p and miR-150-3p were Downregulated in Lung Adenocarcinoma (LUAD) Specimens and Cell Lines

To explore whether *miR-150-5p* and *miR-150-3p* are downregulated in LUAD, expression levels of *miR-150-5p* and *miR-150-3p* in clinical specimens were measured. The characteristics of the patients are recorded in Table 1. 

We found that expression of the *miR-150* duplex was significantly decreased in LUAD specimens and LUAD cell lines relative to non-cancerous tissues (*p* = 0.0078 and *p* < 0.0001, respectively, Figure 1A,B). There were positive correlations between the expression levels of *miR-150-5p* and *miR-150-3p* by Spearman’s rank test (*r* = 0.7156 and *p* < 0.0001, Figure 1C). According to the Kaplan–Meier overall survival curves using TCGA database, we found that the low expression levels of *miR-150-5p* and *miR-150-3p* were associated with poor prognosis in LUAD patients (overall survival, *p* = 0.0078 and *p* = 0.0435, respectively, Figure 1D,E).

### 2.2. Overexpression of miR-150-5p and miR-150-3p Inhibits Cancer Cell Aggressiveness

To investigate the biological functions of *miR-150-5p* and *miR-150-3p* in LUAD, we performed gain-of-function assays using miRNA transfection into LUAD cell lines (A549 and H1299). Cell proliferation assays showed that *miR-150-5p*- and *miR-150-3p*- transfected LUAD cells had reduced cell growth compared with mock- or miR-control-transfected LUAD cells (Figure 2A). Also, cell migratory and invasive abilities were markedly decreased in the LUAD cells transfected with *miR-150-5p* and *miR-150-3p* (Figure 2B,C). 

Furthermore, we performed functional analysis by reducing the concentration of *miR-150-3p* (1 nM and 0.1 nM) transfection into LUAD cells. Our data showed that antitumor functions (inhibition of cancer cell proliferation, migration, and invasion) were observed at 1 nM concentration, although there were no antitumor functions at 0.1 nM concentration on A549 and H1299 cells (Appendix A).

### 2.3. Incorporation of miR-150-5p and miR-150-3p into the RNA-induced silencing complex (RISC) in LUAD Cells

We next performed immunoprecipitation with antibodies targeting Ago2, which plays a pivotal role in the uptake of miRNAs into the RISC (Appendix A). After transfection of A549 cells with *miR-150-5p* and immunoprecipitation by anti-Ago2 antibodies, *miR-150-5p* levels in the immunoprecipitates were significantly higher than those of mock- or miR-control-transfected cells and those of *miR-150-3p*-transfected cells (*p* < 0.01; Appendix A). Similarly, after *miR-150-3p* transfection (10 nM, 1 nM, and 0.1 nM), substantial amounts of *miR-150-3p* were detected in Ago2 immunoprecipitates (*p* < 0.01; Appendix A).

### 2.4. Candidate Target Genes of miR-150-5p and miR-150-3p Regulation in LUAD: Clinical Significance of TNS4, SFXN1, SKA3, and SPOCK1 Expression

Our selection strategy of *miR-150-5p*- and *miR-150-3p*-regulated oncogenic targets is shown in Appendix A. A total of 41 and 26 oncogenic targets regulated by *miR-150-5p* and *miR-150-3p* were identified in LUAD cells (Table 2 and Table 3). 

In this study, we focused on *miR-150-3p*, which is the passenger strand of the *miR-150* duplex and had pronounced anti-tumor function. We examined the relation between the pathogenesis of LUAD and these targets using TCGA database and found four genes (*TNS4*, *SFXN1*, *SKA3*, *SPOCK1*) that were strongly associated with patient outcomes (5-year overall survival, *p* < 0.01, Figure 3A–D). Finally, we focused on *TNS4*, the expression of which was strongly associated with poor prognosis of LUAD patients (disease-free survival: *p* = 0.0213 and overall survival: *p* = 0.0003) among the four genes and validated the effect on LUAD cells.

### 2.5. miR-150-3p Directly Regulated TNS4 in A549 Cells

In cells transfected with *miR-150-3p*, the levels of *TNS4* mRNA and TNS4 protein were significantly lower than mock- or miR-control-transfected cells (Figure 4A,B). Furthermore, the expression of *TNS4* mRNA and TNS4 protein was suppressed at the diluted concentration of *miR-150-3p* precursor (1 nM and 0.1 nM) (Appendix A). 

In order to confirm the binding site of *miR-150-3p*, the nucleotide sequences of 3’UTR (UTR: untranslated region) of *TNS4* in A549 cells was examined independently. Our data showed that several variants of the 3’UTR of *TNS4* existed in A549 cells (Appendix A). As a result of sequencing analyses, one putative binding site of the *miR-150-3p* was found in 3’UTR of *TNS4* (Appendix A). Based on our sequence data, we used luciferase reporter assays with vectors carrying either the wild-type or deletion-type 3’-UTR of *TNS4* (Figure 4C). We observed significantly reduced luminescence intensities after transfection with *miR-150-3p* and the wild-type 3’-UTR of *TNS4* (Figure 4C). Transfection with the deletion-type vector did not reduced luminescence intensities in A549 cells (Figure 4C). Thus, *miR-150-3p* directly bound to *TNS4* in the 3’-UTR. Although TargetScanHuman database predicted putative binding sites of *miR-150-5p* in 3’UTR of *TNS4*, our sequencing data could not confirm the sequences.

### 2.6. Expression of TNS4 Protein in Clinical LUAD Specimens

Analysis using a tissue microarray was performed to examine TNS4 expression at the protein level. We validated the expression of TNS4 by using immunohistochemical staining. In this study, we stained 20 LUAD specimens and 14 non-cancerous specimens. Clinical information on the tissue microarray is shown in the Appendix A. Compared with non-cancerous tissues, TNS4 proteins were highly expressed in LUAD specimens (Figure 5). 

### 2.7. TNS4 Silencing Suppresses the Aggressiveness of LUAD Cells

To confirm the effect of *TNS4* on LUAD cells, we used si-*TNS4* to knock down its expression in A549 cells. RT-PCR and Western blotting showed that expression levels of both *TNS4* mRNA and TNS4 protein were markedly reduced by both si-*TNS4*-1 and si-*TNS4*-2 (Figure 6A,B). In functional assays, cell proliferation, migration, and invasive abilities were significantly suppressed by si-*TNS4* transfection in LUAD cells (Figure 6C–E).

### 2.8. Gain-of-Function Studies by TNS4 Expression Vector

To explore whether *TNS4* promoted cell proliferation, migration, and invasive abilities, we transfected pCMV-*TNS4* into H1299 cells and performed functional assays. Western blotting indicated that overexpression of TNS4 protein was observed in pCMV-*TNS4* vector-transfected cells (Figure 7A). Furthermore, cell migration and invasion were significantly enhanced in pCMV-*TNS4* vector-transfected cells (Figure 7B–D).

### 2.9. Downstream Genes Affected by the Silencing of TNS4 in LUAD Cells

We performed genome-wide gene expression analysis in LUAD cells transfected with si-*TNS4* and in silico analysis to investigate the downstream genes regulated by *TNS4*. A total of 1521 downregulated genes were identified using gene expression analysis in LUAD cells transfected with si-*TNS4*. Among them, we found high expression of genes in the NSCLC clinical expression profiles from the GEO database (GEO accession no: GSE 19188). A total of 88 genes were identified as *TNS4*-modulated genes (Table 4 and Appendix A). 

## 3. Discussion

According to the current concept of miRNA biogenesis, miRNA passenger strands are degraded and have no cellular functions. In contrast to this concept, our miRNA signatures based on RNA sequencing revealed that some passenger strands are aberrantly expressed in cancer tissues [10,16,21,24,25,26,27]. Importantly, functional assays showed that some passenger strands of miRNAs (e.g., *miR-144-5p*, *miR-145-3p*, *miR-139-3p*, *miR-199-3p*, *miR-223-3p,* and *miR-455-5p*) actually acted as anti-tumor miRNAs by controlling cancer-related genes [17,25,28,29,30,31,32,33]. In general theory, passenger strand of miRNAs derived from miRNA-duplex have degraded in cytoplasm and have no function. In fact, the expression of passenger strand of miRNA is overwhelmingly lower than that of guide strand. Is the passenger strand of miRNA actually functional *in vivo*? This is an important issue in miRNA research. Expression levels of *miR-150-3p* were lower (100 ×) than *miR-150-5p* in LUAD cells. Our *in vitro* functional assays showed that antitumor effects are observed even if the transfection concentration of mature *miR-150-3p* is lowered (1 nM and 0.1 nM). Elucidation of functions of passenger strand of miRNA *in vivo* is an important biological theme. The involvement of passenger strands of miRNAs in cancer pathogenesis is an attractive proposal for cancer research. Identification of novel molecules controlled by miRNA (the passenger strand of miRNA duplex) will contribute to the understanding of the oncogenic networks of LUAD. 

In previous studies, tumor suppressive function of *miR-150-5p* (the guide strand) was reported in several cancers [18,22,23]. In contrast to this, very few reports have investigated the functional significance of *miR-150-3p* (the passenger strand) in cancer cells and its controlled cancer-related genes. We have revealed anti-tumor function of *miR-150-3p* in esophageal, head and neck, and lung squamous cell carcinomas [21,22,23]. Moreover, we revealed *miR-150-3p* targets oncogenes involved in the focal adhesion pathways (e.g., *SPOCK1*, *TNC*, *ITGA3,* and *ITGA6*) [21,22,23]. Importantly, these genes were overexpressed in cancer tissues and their high expression was significantly correlated with poor prognosis [21,22,23]. 

In the present study, we finally identified four oncogenes (*TNS4*, *SFXN1*, *SKA3,* and *SPOCK1*) regulated by *miR-150-3p* in LUAD cells. Expression of these genes were significantly associated with LUAD pathogenesis. It is interesting to note that knockdown of *SPOCK1* significantly attenuated cancer cell migration and invasive abilities [21,22,23]. Another target gene, *SKA3,* was overexpressed in renal cell carcinoma and its aberrant expression was associated with cancer cell malignant phenotypes [32]. These data showed that target genes by *miR-150-3p* regulation closely contributed to LUAD pathogenesis and tumorigenesis. Detailed analysis of *miR-150-3p* target genes is indispensable for elucidating the molecular mechanism of LUAD. 

We further investigated the oncogenic roles of *TNS4* (tensin 4) because high expression of *TNS4* was strongly associated with poor prognosis of LUAD patients (disease-free survival: *p* = 0.0213 and overall survival: *p* = 0.0003). *TNS4* (alias *CTEN*) is a member of the tensin family, which includes *TNS1*–*TNS4*. *TNS4* contains SH2 (Src homology 2) and PTB (phosphotyrosine binding) domains at the C-terminal region, both of which are shared with other tensin family members [34]. Interestingly, these domains are essential to bind integrin β1, c-Cbl, β-catenin, and Eepidermal growth factor (EGF) receptor [35,36]. Previous study showed that c-Cbl is an E3 ubiquitin protein ligase and induced EGFR ubiquitination. TNS4 bound to c-Cbl reduced EGFR ubiquitination and degradation [36]. As a result, overexpression of TNS4 stabilized EGFR and enhanced its oncogenic signaling in cancer cells [36]. 

Aberrant expression of *TNS4* was reported in cancers of the breast, colon, pancreas, and lung, and its expression was associated with poorer prognosis of these patients [34,37,38,39]. In hepatocellular carcinoma, EGF-induced extracellular signal-regulated kinase (ERK)1/2 activation enhanced TNS4 expression and these events enhanced the epithelial–mesenchymal transition (EMT) phenotype [40]. In lung cancer, TNS4 was upregulated by EGF-mediated STAT3 activation and its aberrant expression increased cancer cell invasiveness [41]. Our present data confirmed the oncogenic features of *TNS4* in LUAD cells. These finding indicate that expression of *TNS4* might be a good prognostic indicator and a promising therapeutic target for LUAD.

Finally, to investigate *TNS4*-modulated oncogenes in LUAD cells, we applied genome-wide gene expression analyses using knockdown of *TNS4*. A total of 88 genes were identified as putative *TNS4*-modulated targets in LUAD cells. Aberrant expression of nine genes (*C1QTNF6*, *TNS4*, *CDK1*, *SPAG5*, *MKI67*, *CHAF1B*, *ARHGAP11A*, *PRC1*, *RHOV*, *p* < 0.001) was closely associated with poor prognosis of patients with LUAD. Cyclin-dependent kinases (CDKs) are critical regulators of cell cycle progression and related to cancer aggressiveness. *CDK1* is essential for cycle progression during the G2/M transition and mitosis. High expression of *CDK1* is associated with poor prognosis in LUAD [42]. SPAG5 is a microtubule-associated protein and is involved in regulating cell cycle progression [43]. Overexpression of *SPAG5* was observed in NSCLC and promoted cell proliferation and invasion through activation of the Akt signaling pathway [44]. *PRC1*, which acts as an organizing anti-parallel microtubule in the central spindle in cytokinesis, is required for tumorigenesis driven by oncogenic K-RAS and loss of p53 in a mouse model for NSCLC [45]. Thus, the data revealed that many of the genes controlled by anti-tumor *miR-150-3p* and by *TNS* are closely involved in the pathogenesis of cancer. The elucidation of the novel targets controlled by anti-tumor miRNAs will accelerate comprehensive understanding of oncogenic networks of LUAD. 

## 4. Materials and Methods 

### 4.1. Human LUAD Specimens and Cell Lines

In this study, 18 LUAD clinical samples and 28 normal lung samples were obtained from the patients who underwent lung surgery at Kagoshima University Hospital from 2010 to 2013. Table 1 presents the clinical characteristics of these patients. The LUAD samples were staged according to the Association for the Study of Lung Cancer TNM classification, seventh edition. 

We used the two LUAD cell lines: A549 and H1299, purchased from the American Type Culture Collection (Manassas, VA, USA). 

We obtained informed consent from all of the patients. The present study was approved by the Bioethics Committee of Kagoshima University (Kagoshima, Japan; approval no. 26-164).

### 4.2. RNA Extraction and Quantitative Real-Time PCR

We carried out RNA extraction from formalin-fixed, paraffin-embedded specimens and cell lines and quantitative real-time reverse transcription-PCR (qRT-PCR) as previously described [18,46,47,48]. The TaqMan probes and primers were listed in Appendix A.

### 4.3. Transfection of miRNAs, siRNAs, and Plasmid Vectors into LUAD Cells 

Transfection protocol of miRNA or siRNA species into cancer cells was described in our previous studies [18,46,47,48]. The reagents used in this study are listed in Appendix A.

### 4.4. Incorporation of miR-150-5p or miR-150-3p into the RISC by Ago2 Immunoprecipitation

A549 cells were transfected with 10 nM miRNAs by reverse transfection. After 72 h, immunoprecipitation was performed using a microRNA Isolation Kit, Human Ago2 (Wako Pure Chemical Industries, Ltd., Osaka, Japan) as described previously [47,48]. Expression levels of *miR-150-5p* and *miR-150-3p* were analyzed by qRT-PCR. MiRNA data were normalized to expressions of *miR-16-5p*, *miR-21-5p,* and *miR-26a*, which were not affected by *miR-150-5p* and *miR-150-3p*. 

### 4.5. Cell Proliferation, Migration, and Invasion Assays

The procedures for assessing cell proliferation, migration and invasion were described previously [18,46,47,48].

### 4.6. Identification of Putative Target Genes Regulated by miR-150-5p and miR-150-3p in LUAD Cells

We identified putative target genes possessing sequences binding to *miR-150-5p* and *miR-150-3p* from the TargetScanHuman database (http://www.targetscan.org/vert_72/). GEO databases GSE19188 and GSE93290 was used for assessment of the association between target genes and the expression of NSCLC clinical specimens. Our strategy for identification of *miR-150-5p* and *miR-150-3p* target genes is outlined in Appendix A. 

### 4.7. Plasmid Construction and Dual Luciferase Reporter Assay

The following two sequences were cloned into the psiCHECK-2 vector (C8021; Promega, Madison, WI, USA): the wild-type sequence of the 3’ untranslated regions (UTRs) of *TNS4*, or the deletion type, which lacked the *miR-150-3p* target sites from *TNS4*. The procedures for transfection and dual luciferase reporter assays were provided in previous studies [47,48].

### 4.8. Clinical Database Analysis of LUAD

We investigated the clinical significance of miRNAs and their target genes with TCGA (https://tcga-data.nci.nih.gov/tcga/) in LUAD. Gene expression and clinical data were obtained from cBioPortal (http://www.cbioportal.org/) and OncoLnc (http://www.oncolnc.org/) (data downloaded on 28 September 2018) [46,47,48,49]. 

### 4.9. Western Blotting and Immunohistochemistry

The procedures for Western blotting and immunohistochemistry were described previously [18,46,47,48]. A tissue microarray was bought from US Biomax (catalog no: BC04002a; Derwood, MD, USA). Primary antibodies were shown in Appendix A.

### 4.10. Statistical Analysis

To assess the significance of differences between two groups, we used Mann–Whitney *U* tests. Differences between multiple groups were assessed by one-way ANOVA and Tukey tests for post-hoc analysis. Tests utilized GraphPad Prism7 (GraphPad Software, La Jolla, CA, USA) and JMP Pro 14 (SAS Institute Inc., Cary, NC, USA).

## 5. Conclusions

Our results showed that the expression of both strands of the pre*-miR-150* duplex was significantly downregulated in LUAD clinical specimens and the *miR-150* duplex acted as an anti-tumor miRNA in LUAD cells. Involvement of the passenger strand of miRNA in LUAD oncogenesis suggests new possible mechanisms of pathogenesis. A total of 26 genes regulated by *miR-150-3p* were closely associated with LUAD pathogenesis. Among these targets, aberrant expression of *TNS4* enhanced cancer aggressiveness, suggesting that *TNS4* could be a promising therapeutic target for LUAD.

## Figures and Tables

**Figure 1 cancers-11-00601-f001:**
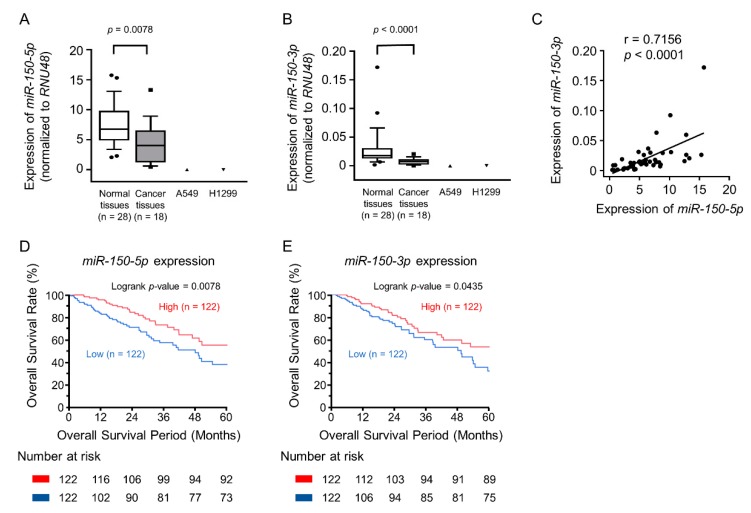
The clinical significance of *miR-150-5p* and *miR-150-3p* expression in lung adenocarcinoma (LUAD). (**A**,**B**) Downregulation of *miR-150-5p* and *miR-150-3p* expression in clinical specimens of LUAD and two cell lines (A549 and H1299). Expression of *RNU48* was used as an internal control. (**C**) Expression of two miRNAs derived from *miR-150*-duplex were positive correlation by Spearman’s rank test. (**D**,**E**) The Kaplan–Meier overall survival curve analyses of patients with LUAD by The Cancer Genome Atlas (TCGA) database. Patients were divided into two groups according to miRNA expression and analyzed.

**Figure 2 cancers-11-00601-f002:**
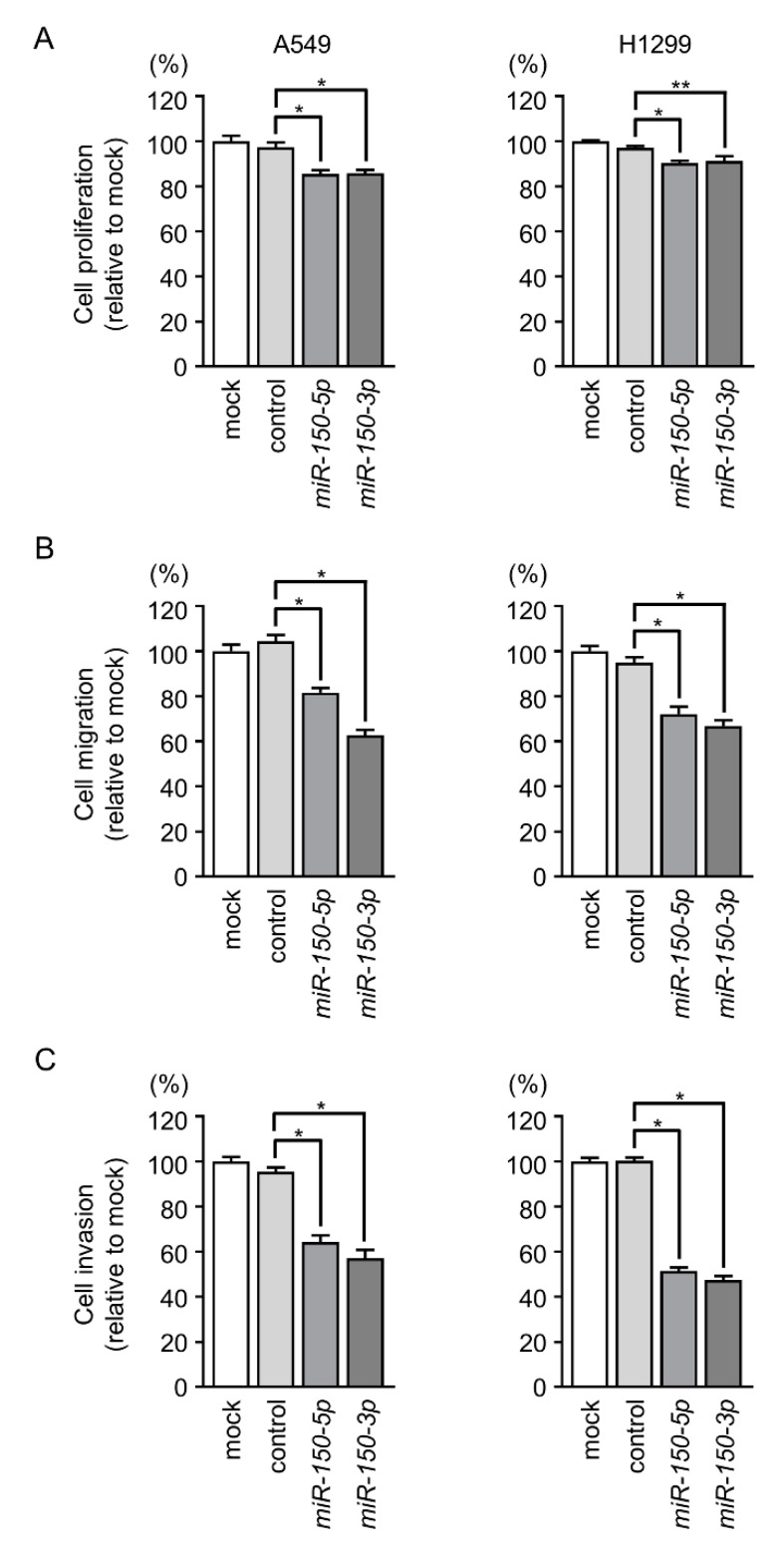
Functional assays of *miR-150-5p* and *miR-150-3p* in LUAD cells (A549 and H1299). (**A**–**C**) Cell proliferation, migration, and invasive activities were significantly blocked by ectopic expression of *miR-150-5p* or *miR-150-3p*. * *p* < 0.01, ** *p* < 0.05.

**Figure 3 cancers-11-00601-f003:**
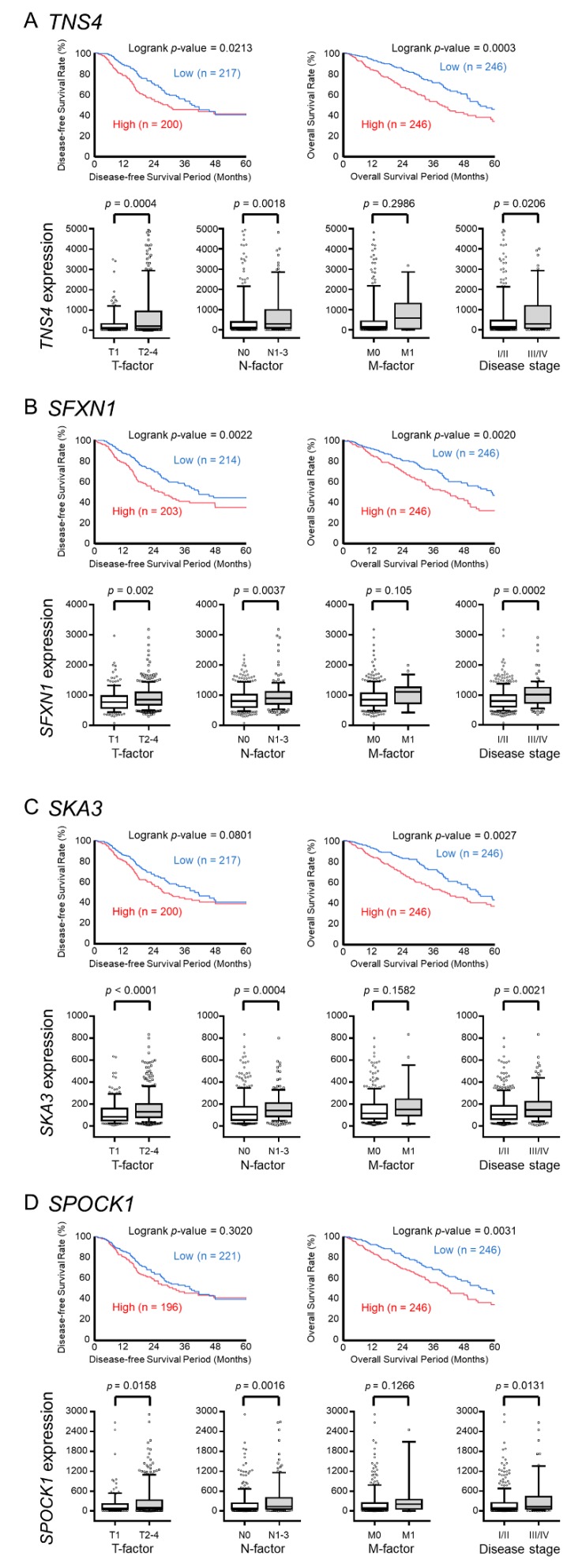
The relationship between the expression levels of four genes (*TNS4*, *SFXN1*, *SKA3*, and *SPOCK1*) and clinical significance based on The Cancer Genome Atlas (TCGA) database. (**A**–**D**) The Kaplan–Meier disease-free survival curves and overall survival curves, T factor, N factor, M factor, and disease stage of the high- and low-expression groups for four genes (*TNS4*, *SFXN1*, *SKA3*, and *SPOCK1*).

**Figure 4 cancers-11-00601-f004:**
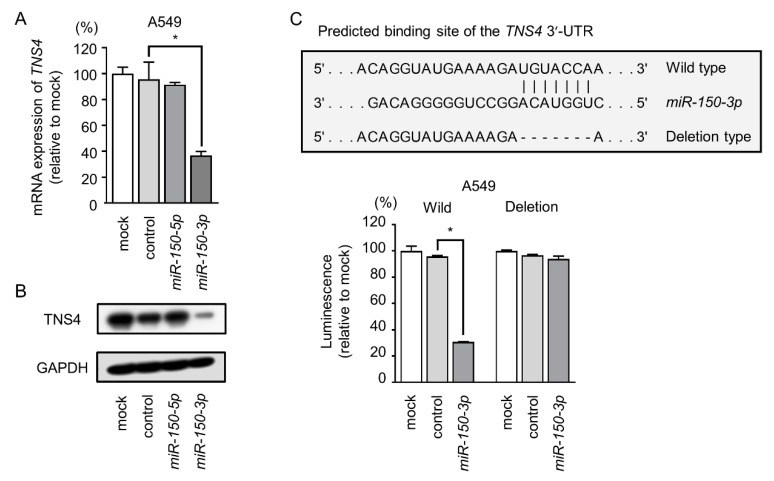
*TNS4* was directly controlled by *miR-150-3p* in LUAD cells. (**A**,**B**) *TNS4* mRNA and protein expression was reduced by *miR-150-3p* ectopic expression (48 h after transfection). *GUSB* was used as an expression control. GAPDH was used as a loading control. * *p* < 0.01. (**C**) Dual luciferase reporter assays using vectors encoding the wild-type *TNS4* 3′-UTR sequence containing one putative *miR-150-3p* target site (wild) and 3′-UTR sequences with deletions of the target site (deletion). Normalized data were calculated as the ratio of *Renilla*/firefly luciferase activities. * *p* < 0.01. UTR: untranslated region.

**Figure 5 cancers-11-00601-f005:**
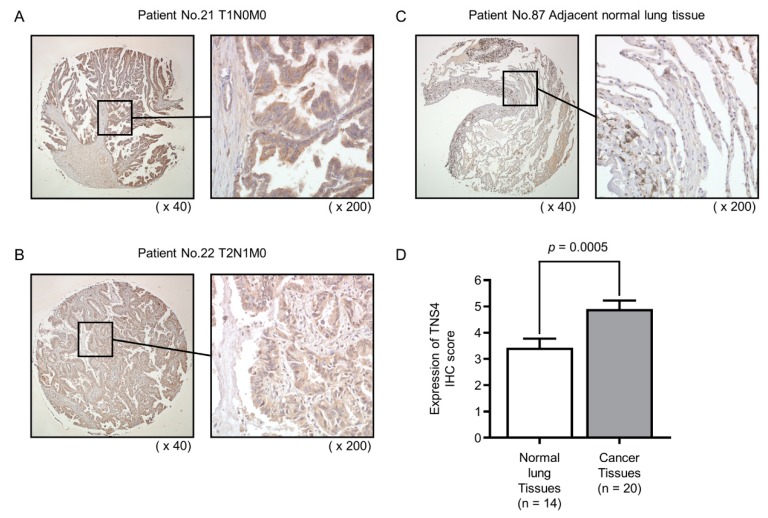
Immunohistochemical staining of TNS4 protein in clinical LUAD specimens. (**A**,**B**) The overexpression of TNS4 was observed in the cytoplasm of cancer cells. (**C**) TNS4 was weakly stained or not detected in normal lung specimens. (**D**) Comparison of immunohistochemical staining of TNS4 in LUAD specimens and normal lung specimens. LUAD specimens showed higher expression of TNS4 than normal lung specimens.

**Figure 6 cancers-11-00601-f006:**
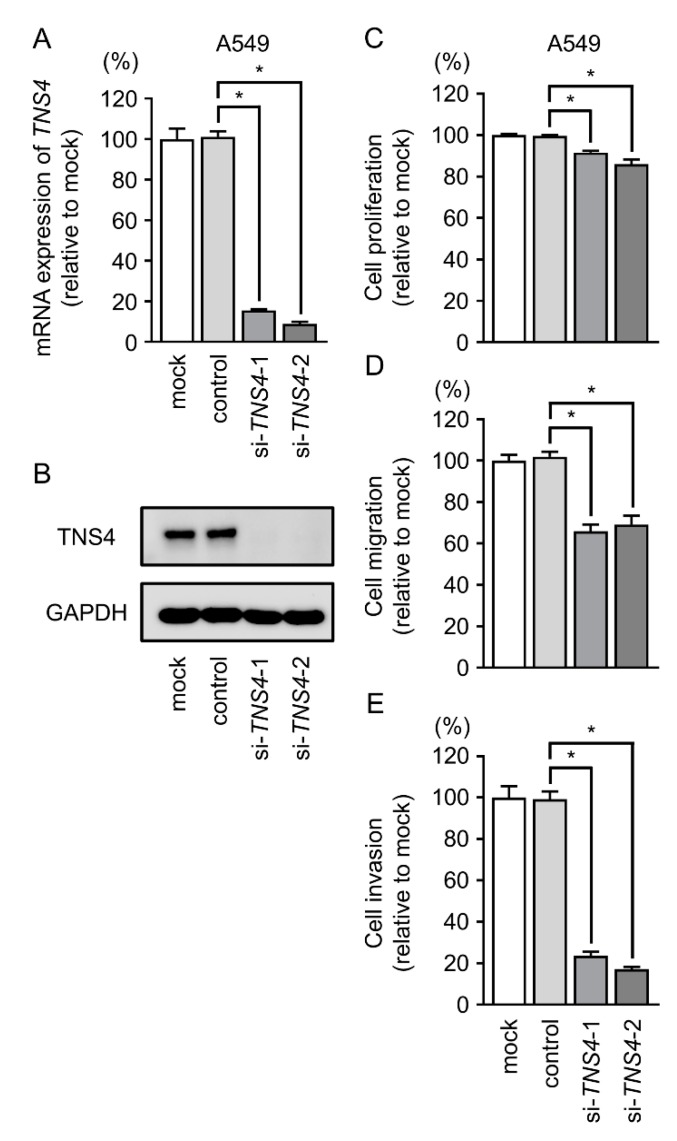
Knockdown studies of *TNS4*/TNS4 using si-*TNS4* in LUAD cells (A549 and H1299). (**A**,**B**) *TNS4* mRNA and protein expression 48 h after transfection of si-*TNS4*-1 or si-*TNS4*-2 in LUAD cell lines. (**C**–**E**) Cell proliferation, migration, and invasive activities were significantly blocked by si-*TNS4* transfection into LUAD cell lines. * *p* < 0.01.

**Figure 7 cancers-11-00601-f007:**
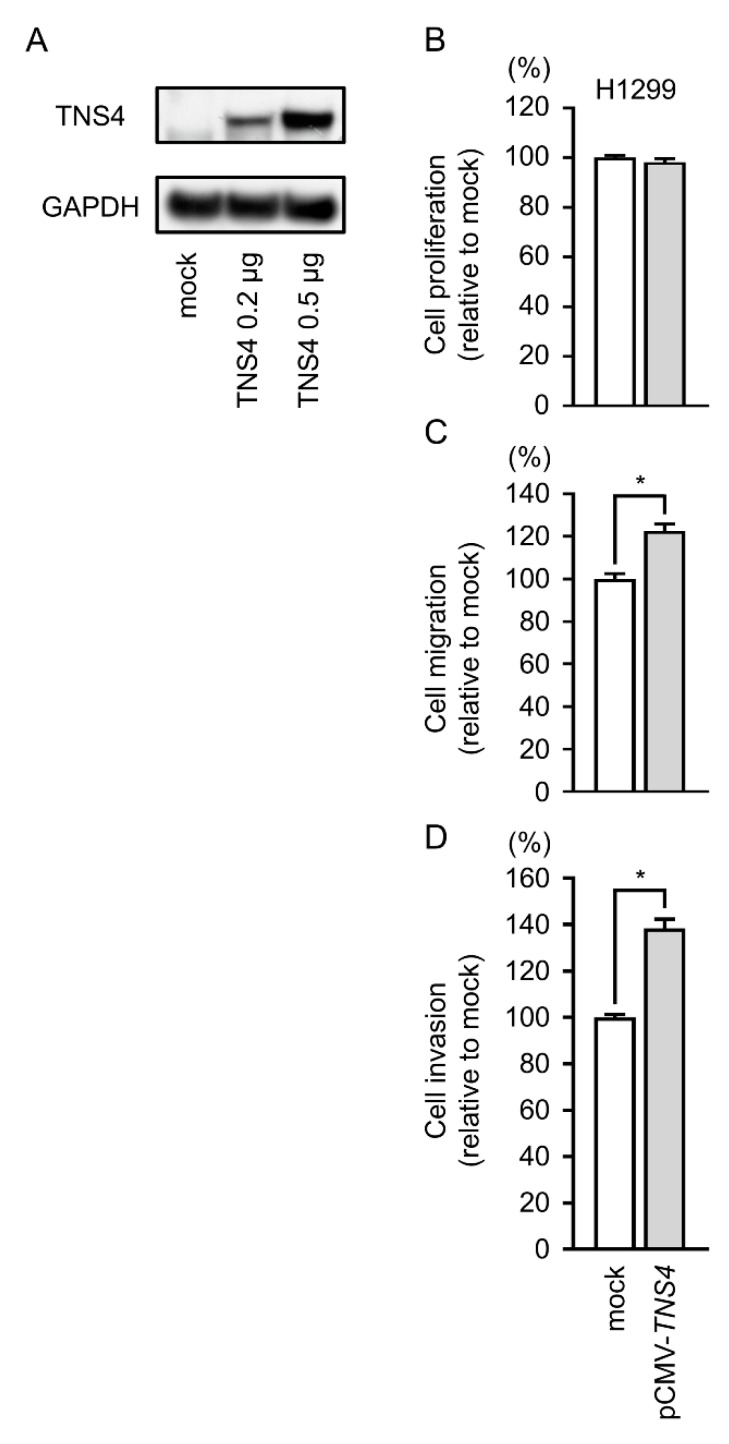
The effects of *TNS4* overexpression by H1299 cells. (**A**) Expression levels of TNS4 protein 48 h after transfection with pCMV-*TNS4* vector into H1299 cells. (**B**) Cell proliferation was evaluated by XTT assays. (**C**) Cell migration was determined by wound healing assay 24 h after forward transfection with the pCMV-*TNS4* vector. (**D**) Cell invasion was determined by Matrigel assay 24 h after forward transfection with the pCMV-*TNS4* vector. These assays showed that migratory and invasive activities were increased in pCMV-*TNS4* vector-transfected cell lines. * *p* < 0.01.

**Table 1 cancers-11-00601-t001:** Characteristics of lung cancer and non-cancerous cases.

**A. Characteristics of the Lung Cancer Cases**		
**Lung Cancer Patients**	**n**	**(%)**
Total number	18	
Median age (range)	73.5 (59–86)	
Gender		
Male	9	50.0
Female	9	50.0
Pathological stage:		
IA	1	5.6
IB	4	22.2
IIA	8	44.4
IIB	1	5.6
IIIA	4	22.2
IIIB	0	0.0
**B. Characteristics of the Non-Cancerous Cases**		
**Non-Cancerous Tissues**	**n**	**(%)**
Total number	28	
Median age (range)	71 (50–88)	
Gender:		
Male	25	89.3
Female	3	10.7

**Table 2 cancers-11-00601-t002:** Candidate target genes regulated by *miR-150-5p.*

Entrez Gene	Gene Symbol	Gene Name	Total Sites	A549 *miR-150-5p* Transfectant FC (log_2_)	GSE19188 FC (log_2_)	TCGA OncoLnc *p*-Value
4751	*NEK2*	NIMA-related kinase 2	1	−0.565	3.323	<0.0001
64065	*PERP*	PERP, TP53 apoptosis effector	3	−3.012	1.835	<0.0001
5122	*PCSK1*	Proprotein convertase subtilisin/kexin type 1	1	−0.680	2.532	0.0006
84985	*FAM83A*	Family with sequence similarity 83, member A	1	−0.717	3.188	0.0010
1033	*CDKN3*	Cyclin-dependent kinase inhibitor 3	2	−0.582	2.889	0.0011
6241	*RRM2*	Ribonucleotide reductase M2	3	−2.433	3.000	0.0013
79801	*SHCBP1*	SHC SH2-domain binding protein 1	1	−1.118	1.841	0.0015
29127	*RACGAP1*	Rac GTPase activating protein 1	2	−0.545	1.677	0.0019
24137	*KIF4A*	Kinesin family member 4A	1	−0.933	3.309	0.0030
6695	*SPOCK1*	Pparc/osteonectin, cwcv and kazal-like domains proteoglycan (testican) 1	1	−0.764	1.696	0.0031
9837	*GINS1*	GINS complex subunit 1 (Psf1 homolog)	2	−0.749	2.991	0.0076
339761	*CYP27C1*	Cytochrome P450, family 27, subfamily C, polypeptide 1	4	−2.360	1.706	0.0126
10331	*B3GNT3*	UDP-GlcNAc:betaGal beta-1,3-N-acetylglucosaminyltransferase 3	2	−0.939	1.608	0.0129
79962	*DNAJC22*	DnaJ (Hsp40) homolog, subfamily C, member 22	5	−0.908	2.031	0.0201
10635	*RAD51AP1*	RAD51-associated protein 1	1	−0.611	2.470	0.0228
10797	*MTHFD2*	Methylenetetrahydrofolate dehydrogenase (NADP+ dependent) 2, methenyltetrahydrofolate cyclohydrolase	1	−1.905	1.887	0.0236
9699	*RIMS2*	Regulating synaptic membrane exocytosis 2	2	−0.697	1.976	0.0265
1058	*CENPA*	Centromere protein A	1	−1.019	3.488	0.0365
23657	*SLC7A11*	Solute carrier family 7 (anionic amino acid transporter light chain, xc- system), member 11	3	−1.093	2.014	0.0474
3755	*KCNG1*	Potassium voltage-gated channel, subfamily G, member 1	5	−1.069	1.887	0.0557
1825	*DSC3*	Desmocollin 3	1	−1.247	2.488	0.0667
388228	*SBK1*	SH3 domain binding kinase 1	1	−0.658	1.745	0.0939
8038	*ADAM12*	ADAM metallopeptidase domain 12	2	−0.668	2.753	0.1097
84733	*CBX2*	Chromobox homolog 2	1	−0.591	1.988	0.1116
130827	*TMEM182*	Transmembrane protein 182	1	−0.690	1.568	0.1189
92312	*MEX3A*	mex-3 RNA binding family member A	3	−0.584	1.986	0.1271
4323	*MMP14*	Matrix metallopeptidase 14	3	−0.621	1.872	0.1296
4151	*MB*	myoglobin	3	−1.119	1.614	0.1346
57167	*SALL4*	Sal-like 4 (Drosophila)	1	−0.582	2.836	0.2382
256714	*MAP7D2*	MAP7 domain containing 2	1	−1.449	2.007	0.2798
6273	*S100A2*	S100 calcium binding protein A2	1	−0.643	2.513	0.3132
200844	*C3orf67*	Chromosome 3 open reading frame 67	1	−0.514	1.584	0.3482
1690	*COCH*	Cochlin	2	−0.592	3.406	0.3696
10447	*FAM3C*	Family with sequence similarity 3, member C	1	−1.570	1.536	0.3847
147920	*IGFL2*	IGF-like family member 2	1	−0.719	2.569	0.4596
547	*KIF1A*	Kinesin family member 1A	4	−0.733	2.518	0.6354
9066	*SYT7*	Synaptotagmin VII	2	−0.921	1.730	0.7141
55220	*KLHDC8A*	Kelch domain containing 8A	4	−1.168	1.709	0.7484
440590	*ZYG11A*	Zyg-11 family member A, cell cycle regulator	2	−1.619	1.826	0.7530
3141	*HLCS*	Holocarboxylase synthetase	1	−0.897	1.791	0.8947
9547	*CXCL14*	Chemokine (C-X-C motif) ligand 14	2	−0.750	1.800	0.9229

**Table 3 cancers-11-00601-t003:** Candidate target genes regulated by *miR-150-3p.*

Entrez Gene	Gene Symbol	Gene Name	Total Sites	A549 *miR-150-3p* Transfectant FC (log_2_)	GSE19188 FC (log_2_)	TCGA OncoLnc *p*-Value
84951	*TNS4*	Tensin 4	1	−1.318	2.560	0.0003
94081	*SFXN1*	Sideroflexin 1	1	−1.307	1.404	0.0020
221150	*SKA3*	Spindle and kinetochore-associated complex subunit 3	1	−1.006	2.015	0.0027
6695	*SPOCK1*	Sparc/osteonectin, cwcv and kazal-like domains proteoglycan (testican) 1	3	−2.040	1.696	0.0031
89874	*SLC25A21*	Solute carrier family 25 (mitochondrial oxoadipate carrier), member 21	2	−2.557	1.358	0.0149
94032	*CAMK2N2*	Calcium/calmodulin-dependent protein kinase II inhibitor 2	1	−1.303	1.538	0.0208
5738	*PTGFRN*	Prostaglandin F2 receptor inhibitor	2	−1.750	1.237	0.0291
23105	*FSTL4*	Follistatin-like 4	1	−1.602	1.485	0.0589
130574	*LYPD6*	LY6/PLAUR domain containing 6	1	−1.004	1.917	0.0591
150223	*YDJC*	YdjC homolog (bacterial)	1	−1.592	1.088	0.0699
3174	*HNF4G*	Hepatocyte nuclear factor 4, gamma	1	−1.827	1.709	0.0717
6857	*SYT1*	Synaptotagmin I	1	−2.264	2.058	0.0735
5522	*PPP2R2C*	Protein phosphatase 2, regulatory subunit B, gamma	2	−1.004	2.491	0.1067
8038	*ADAM12*	ADAM metallopeptidase domain 12	2	−1.555	2.753	0.1097
84216	*TMEM117*	Transmembrane protein 117	1	−3.167	1.049	0.1894
55753	*OGDHL*	Oxoglutarate dehydrogenase-like	1	−1.896	1.776	0.3167
79944	*L2HGDH*	L-2-hydroxyglutarate dehydrogenase	1	−1.921	1.730	0.3221
79776	*ZFHX4*	Zinc finger homeobox 4	1	−2.113	1.248	0.3578
4647	*MYO7A*	Myosin VIIA	1	−1.193	1.047	0.4080
23321	*TRIM2*	Tripartite motif containing 2	2	−1.550	1.755	0.4560
401474	*SAMD12*	Sterile alpha motif domain containing 12	1	−1.950	1.027	0.4853
145282	*MIPOL1*	Mirror-image polydactyly 1	1	−1.401	1.108	0.5861
8821	*INPP4B*	Inositol polyphosphate-4-phosphatase, type II, 105kDa	3	−1.412	1.414	0.6077
80310	*PDGFD*	Platelet-derived growth factor D	1	−1.555	1.142	0.6859
9802	*DAZAP2*	DAZ-associated protein 2	2	−1.439	1.144	0.8241
85439	*STON2*	Stonin 2	1	−1.371	1.079	0.9773

**Table 4 cancers-11-00601-t004:** Candidate downstream genes modulated by *TNS4* in lung adenocarcinoma.

Entrez Gene	Gene Symbol	Gene Name	GSE19188 FC (log_2_)	A549 si-*TNS4*-2 Transfectant FC (log_2_)	TCGA OncoLnc *p*-Value
114904	*C1QTNF6*	C1q and tumor necrosis factor related protein 6	2.046	−1.159	<0.0001
983	*CDK1*	Cyclin-dependent kinase 1	2.400	−1.119	0.0003
10615	*SPAG5*	Sperm-associated antigen 5	2.196	−1.598	0.0003
84951	*TNS4*	Tensin 4	2.560	−2.690	0.0003
4288	*MKI67*	Marker of proliferation Ki-67	2.835	−1.475	0.0004
8208	*CHAF1B*	Chromatin assembly factor 1, subunit B (p60)	1.723	−1.112	0.0005
9824	*ARHGAP11A*	Rho GTPase activating protein 11A	1.638	−1.263	0.0007
9055	*PRC1*	Protein regulator of cytokinesis 1	2.540	−1.120	0.0007
171177	*RHOV*	Ras homolog family member V	2.330	−1.890	0.0008
1033	*CDKN3*	Cyclin-dependent kinase inhibitor 3	2.889	−1.594	0.0011
6241	*RRM2*	Ribonucleotide reductase M2	3.000	−1.356	0.0013
57405	*SPC25*	SPC25, NDC80 kinetochore complex component	2.417	−1.456	0.0014
5318	*PKP2*	Plakophilin 2	1.584	−1.108	0.0016
701	*BUB1B*	BUB1 mitotic checkpoint serine/threonine kinase B	2.669	−1.278	0.0017
4085	*MAD2L1*	MAD2 mitotic arrest deficient-like 1 (yeast)	2.768	−1.726	0.0018
81624	*DIAPH3*	Diaphanous-related formin 3	1.926	−1.016	0.0022
3832	*KIF11*	Kinesin family member 11	2.479	−1.007	0.0022
79019	*CENPM*	Centromere protein M	2.391	−1.057	0.0023
55635	*DEPDC1*	DEP domain containing 1	3.443	−1.282	0.0024
147841	*SPC24*	SPC24, NDC80 kinetochore complex component	2.179	−1.490	0.0031
195828	*ZNF367*	Zinc finger protein 367	1.583	−1.454	0.0033
1063	*CENPF*	Centromere protein F, 350/400kDa	2.985	−1.129	0.0048
83540	*NUF2*	NUF2, NDC80 kinetochore complex component	3.442	−1.069	0.0048
29089	*UBE2T*	Ubiquitin-conjugating enzyme E2T	3.317	−1.092	0.0051
7348	*UPK1B*	Uroplakin 1B	1.603	−1.179	0.0059
11130	*ZWINT*	ZW10 interacting kinetochore protein	2.184	−1.188	0.0064
11169	*WDHD1*	WD repeat and HMG-box DNA binding protein 1	2.094	−1.151	0.0065
55215	*FANCI*	Fanconi anemia, complementation group I	2.298	−1.563	0.0073
4176	*MCM7*	Minichromosome maintenance complex component 7	1.555	−1.164	0.0076
10403	*NDC80*	NDC80 kinetochore complex component	2.493	−1.053	0.0076
699	*BUB1*	BUB1 mitotic checkpoint serine/threonine kinase	3.206	−1.180	0.0081
79075	*DSCC1*	DNA replication and sister chromatid cohesion 1	1.704	−1.187	0.0088
51514	*DTL*	Denticleless E3 ubiquitin protein ligase homolog (Drosophila)	2.098	−1.187	0.0091
8914	*TIMELESS*	Timeless circadian clock	1.650	−1.447	0.0093
79733	*E2F8*	E2F transcription factor 8	2.879	−2.313	0.0114
4605	*MYBL2*	v-Myb avian myeloblastosis viral oncogene homolog-like 2	3.003	−1.063	0.0122
5427	*POLE2*	Polymerase (DNA directed), epsilon 2, accessory subunit	1.600	−1.074	0.0122
10331	*B3GNT3*	UDP-GlcNAc:betaGal beta-1,3-N-acetylglucosaminyltransferase 3	1.608	−1.005	0.0129
7083	*TK1*	Thymidine kinase 1, soluble	2.086	−1.399	0.0131
6790	*AURKA*	Aurora kinase A	2.542	−1.018	0.0132
79623	*GALNT14*	Polypeptide N-acetylgalactosaminyltransferase 14	2.572	−1.075	0.0140
64151	*NCAPG*	Non-SMC condensin I complex, subunit G	2.833	−2.675	0.0147
8318	*CDC45*	Cell division cycle 45	3.829	−1.605	0.0159
55165	*CEP55*	Centrosomal protein 55kDa	2.875	−1.077	0.0203
54478	*FAM64A*	Family with sequence similarity 64, member A	2.713	−1.697	0.0219
79172	*CENPO*	Centromere protein O	1.620	−1.048	0.0248
2491	*CENPI*	Centromere protein I	2.088	−1.685	0.0258
1356	*CP*	Ceruloplasmin (ferroxidase)	1.604	−2.134	0.0300
2244	*FGB*	Fibrinogen beta chain	1.887	−3.066	0.0305
29128	*UHRF1*	Ubiquitin-like with PHD and ring finger domains 1	2.576	−1.442	0.0312
1058	*CENPA*	Centromere protein A	3.488	−1.338	0.0365
2877	*GPX2*	Glutathione peroxidase 2 (gastrointestinal)	3.579	−1.038	0.0446
5888	*RAD51*	RAD51 recombinase	2.085	−1.344	0.0478
8438	*RAD54L*	RAD54-like (S. cerevisiae)	2.920	−1.013	0.0509
1789	*DNMT3B*	DNA (cytosine-5-)-methyltransferase 3 beta	1.606	−1.029	0.0607
5984	*RFC4*	Replication factor C (activator 1) 4, 37kDa	2.004	−1.166	0.0938
91057	*CCDC34*	Coiled-coil domain containing 34	1.995	−1.985	0.1035
202915	*TMEM184A*	Transmembrane protein 184A	1.940	−1.983	0.1067
8038	*ADAM12*	ADAM metallopeptidase domain 12	2.753	−1.464	0.1097
51557	*LGSN*	Lengsin, lens protein with glutamine synthetase domain	1.721	−1.008	0.1253
4151	*MB*	Myoglobin	1.614	−2.307	0.1346
201299	*RDM1*	RAD52 motif containing 1	1.503	−1.596	0.1583
5080	*PAX6*	Paired box 6	1.602	−1.692	0.1649
114907	*FBXO32*	F-box protein 32	1.990	−1.044	0.1654
286151	*FBXO43*	F-box protein 43	1.569	−1.906	0.1729
10293	*TRAIP*	TRAF interacting protein	2.362	−1.114	0.1742
83990	*BRIP1*	BRCA1 interacting protein C-terminal helicase 1	2.051	−1.439	0.1829
349136	*WDR86*	WD repeat domain 86	1.653	−1.306	0.1904
1870	*E2F2*	E2F transcription factor 2	1.704	−1.075	0.1993
3007	*HIST1H1D*	Histone cluster 1, H1d	1.568	−1.265	0.2918
6676	*SPAG4*	Sperm-associated antigen 4	1.552	−1.003	0.3275
200844	*C3orf67*	Chromosome 3 open reading frame 67	1.584	−1.436	0.3482
57016	*AKR1B10*	Aldo-keto reductase family 1, member B10 (aldose reductase)	3.628	−2.490	0.3591
1719	*DHFR*	Dihydrofolate reductase	1.538	−1.033	0.4901
8581	*LY6D*	Lymphocyte antigen 6 complex, locus D	2.171	−1.384	0.5123
6518	*SLC2A5*	Solute carrier family 2 (facilitated glucose/fructose transporter), member 5	1.986	−1.018	0.5410
10535	*RNASEH2A*	Ribonuclease H2, subunit A	1.751	−1.020	0.5517
10018	*BCL2L11*	BCL2-like 11 (apoptosis facilitator)	1.592	−1.310	0.5755
25837	*RAB26*	RAB26, member RAS oncogene family	2.112	−1.420	0.5804
57834	*CYP4F11*	Cytochrome P450, family 4, subfamily F, polypeptide 11	1.795	−1.032	0.6899
100133941	*CD24*	CD24 molecule	2.092	−1.777	0.7765
56521	*DNAJC12*	DnaJ (Hsp40) homolog, subfamily C, member 12	2.208	−1.687	0.7922
3141	*HLCS*	Holocarboxylase synthetase	1.791	−1.020	0.8947
222962	*SLC29A4*	Solute carrier family 29 (equilibrative nucleoside transporter), member 4	1.617	−1.100	0.9072
1645	*AKR1C1*	Aldo-keto reductase family 1, member C1	2.257	–1.546	0.9583
152404	*IGSF11*	Immunoglobulin superfamily, member 11	1.590	–1.572	0.9790
85285	*KRTAP4-1*	Keratin-associated protein 4-1	2.215	–1.029	no data
25859	*PART1*	Prostate androgen-regulated transcript 1 (non-protein coding)	1.915	–1.383	no data

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
