# Peer review of "Molecular Pathogenesis of Gene Regulation by the miR-150 Duplex: miR-150-3p Regulates TNS4 in Lung Adenocarcinoma"

_cancers, 2019, doi:10.3390/cancers11050601_

Round 1

Reviewer 1 Report

The manuscript entitled „Molecular pathogenesis of gene regulation by the miR-150 duplex: miR-150-3p regulates TNS4 in lung adenocarcinoma“ by Misono and colleagues covers the role of miR-150-3p in the disease progression of lung adenocarcinoma. The author describe the passenger strand of the miR-150 duplex to be downregulated in the analyzed LUAD patients samples in comparison to non-cancerous tissues, analogous to the guide strand miR-150-5p. Furthermore, the authors identify several putatively miR-150-3p regulated target genes associated with LUAD progression and confirm tensin 4 (TNS4) as miR-150-3p target gene by luciferase reporter assay and western blot. Finally, the authors demonstrate TNS4 overexpression to enhance the LUAD aggressiveness, and identify 88 potentially TNS4-regulated genes, many of them associated to LUAD disease progression. The manuscript is mostly written understandable, and covers of course an extremely relevant topic in oncology. I also appreciate the thoughtful design of the experiments and the stringent presentation of the results by the authors. However, I have some major concerns on this manuscript:

1) Regarding the number of samples analyzed initially: 18 LUAD specimens is a rather small number, given the high frequency of this tumor entity. Why was this small sample size chosen?

2) Regarding Figure 1: The authors should acknowledge the fact that in their analyzed tissue samples the passenger strand miR-150-3p is 100 – 1,000fold lower expressed as the guide strand miR-150-3p (See Figure 1A vs 1B). However, I do not see that this result is considered in any of the following experiments. For instance, in chapter 2.2. the same concentration of miR-150-5p and miR-150-3p (10 nM) was used to study (and compare) the effects of the overexpression on cancer cell aggressiveness. Although this procedure may be sufficient for the analysis of the cellular reaction on the individual miR species, it does not reflect the biological reality. Therefore, I would recommend to study the effects of lower miR-150-3p concentrations (10 – 100fold lower) on cell proliferation, migration and invasion in comparison.

3) Regarding the miR-150-3p target gene selection: why was SFXN1 not chosen additionally? It exhibits similar prognostic capacity as TNS4, is considered as survival-associated in gliomas (Weston et al., PLoS One 2016) and is rarely studied in LUAD.

4) Regarding the miR-150-3p candidate gene TNS4: then rechecking in TargetScan (release 7.2; March 2018), I was also given 4 poorly conserved binding sites for miR-150-5p (3x 7mer-m8 and 1x 7mer-A1, additionally 2x 6mer sites). Furthermore, miRTarBase (http://mirtarbase.mbc.nctu.edu.tw/php/search.php?opt=search_box&kw=tns4&sort=id&order=asc&page=3) also lists TNS4 as miR-150-5p target, at least in a NGS study. The authors should re-check, whether TNS4 is also a miR-150-5p target gene (I see that the authors controlled in Figure 4, however, I cannot ignore the fact that the initial in silico search also retrieves miR-150-5p binding sites). In this case, I also point to the fact, that the regulation by 4 putative target sites (miR-150-5p) may be much stronger than by one target site (miR-150-3p), overriding the proposed miR-150-3p effect on this oncogene.

5) Regarding Figure 4: the authors demonstrate the regulation of TNS4 by miR-150-3p well, but do not acknowledge the fact that the miR-150-3p binding site is at the end of the around 4600 nt long TNS4 3’UTR. Tumors are known for sometimes shortening some mRNAs 3’UTRs to secure them of posttranscriptional control. Did the authors check for the TNS4-3’UTR lengths in LUAD, for instance by qPCRs using differently located primers on the 3’UTR?

6) Regarding chapter 2.6: were the LUAD specimens analyzed by immunohistochemistry corresponding to that analyzed for miR-150 expression in chapter 2.1.? Because in this chapter, 20 LUAD specimens were analyzed ,compared to 18 in chapter 2.1. Please clarify the difference.

7) There are several sentences scattered over the text, which are rather unspecific. For instance: l.26-28, l.201-203, l.250-252. I suggest to sharpen such rather broad, sometimes – I apologize -  exaggerating statements.

8) Minor issue: in the introduction, the term “contributed to” is sometimes used false (l.47, l.50, etc.) – please correct.

9) Minor issue:typo in l. 112: transfected.

10) Minor issue: the Material and Methods section is quite short. Maybe the authors may give some more details to the reader rather than only redirect to their previous publications.

Finally, I want to thank the authors for sharing their work with the scientific community and apologize for the rather harsh assessment of their work. Best regards.

Author Response

Cancers-467826

Cancers: Special Issue "Molecular Profiling of Lung Cancer"

Revised letter

April 10, 2019

Dr. Kentaro Inamura

Guest Editor

Ms. Diana Wang

Assistant Editor

Dear Dr. Inamura:

We would like to express our gratitude for your consideration of our above-mentioned manuscript for publication in Cancers. Enclosed, please find the revised manuscript Cancers (Cancers-467826) along with a detailed explanation of the revisions, which were made based on the reviewers’ comments. All changes are highlighted in the revised manuscript.

Reviewer #1

Comment-1: Regarding the number of samples analyzed initially: 18 LUAD specimens is a rather small number, given the high frequency of this tumor entity. Why was this small sample size chosen?

Response: Unfortunately, the number of lung cancer samples available to our group is limited. In this experiment, all samples that can currently be analyzed were used. I appreciate your understanding.

Comment-2: Regarding Figure 1: The authors should acknowledge the fact that in their analyzed tissue samples the passenger strand miR-150-3p is 100 – 1,000fold lower expressed as the guide strand miR-150-3p (See Figure 1A vs 1B). However, I do not see that this result is considered in any of the following experiments. For instance, in chapter 2.2. the same concentration of miR-150-5p and miR-150-3p (10 nM) was used to study (and compare) the effects of the overexpression on cancer cell aggressiveness. Although this procedure may be sufficient for the analysis of the cellular reaction on the individual miR species, it does not reflect the biological reality. Therefore, I would recommend to study the effects of lower miR-150-3p concentrations (10 – 100fold lower) on cell proliferation, migration and invasion in comparison.

Response: Reviewer’s comment is an important issue for this paper. In response to the referee's comments, I conducted the following experiment. Functional analysis was carried out by transfection of miR-150-3p (10nM, 1nM and 0.1nM) into LUAD cells. New analysis data is shown in the Supplemental Figure and add the following sentence.

Furthermore, we performed functional analysis by reducing the concentration of miR-150-3p (1nM and 0.1nM) transfection into LUAD cells. Our data showed that antitumor functions (inhibition of cancer cell proliferation, migration and invasion) were observed at 1nM concentration on A549 and H1299 cells (Supplemental Figure 1).

Comment-3: Regarding the miR-150-3p target gene selection: why was SFXN1 not chosen additionally? It exhibits similar prognostic capacity as TNS4, is considered as survival-associated in gliomas (Weston et al., PLoS One 2016) and is rarely studied in LUAD.

Response: I thank you very much for pointing that out.

Recent study showed that SFXN1 (sideroflexin 1) functioned as a mitochondrial serine transporter in one-carbon metabolism (Science, 16;362, 2018). In cancer research, aberrant expression of iron regulatory genes, including SFXN1 associated with poor prognosis of the patients with gliomas, as the reviewer pointed out. However, functional analysis in cancer cells is not enough. Therefore, we are currently analyzing the functional analysis of SFXN1 using several types of cancer cells. In this paper, we will skip this gene. I appreciate your understanding.

Comment-4: Regarding the miR-150-3p candidate gene TNS4: then rechecking in TargetScanHuman (release 7.2; March 2018), I was also given 4 poorly conserved binding sites for miR-150-5p (3x 7mer-m8 and 1x 7mer-A1, additionally 2x 6mer sites). Furthermore, miRTarBase (http://mirtarbase.mbc.nctu.edu.tw/php/search.php?opt=search_box&kw=tns4&sort=id&order=asc&page=3) also lists TNS4 as miR-150-5p target, at least in a NGS study. The authors should re-check, whether TNS4 is also a miR-150-5p target gene (I see that the authors controlled in Figure 4, however, I cannot ignore the fact that the initial in silico search also retrieves miR-150-5p binding sites). In this case, I also point to the fact, that the regulation by 4 putative target sites (miR-150-5p) may be much stronger than by one target site (miR-150-3p), overriding the proposed miR-150-3p effect on this oncogene.

Response: I thank you very much for pointing that out. As suggested by the reviewer’s comment, I re-analyzed the regulation of TNS4 by miR-150-5p (guide strand) in cancer cells.

We investigated the several types of cancer cells, e.g., lung squamous cell carcinoma, head and neck cancer, esophageal cancer, prostate cancer. Our data showed that there was no significant suppression of TNS4 expression by miR-150-5p transfection.

We checked the length of 3’UTR of TNS4 in cancer cells, and we obtained data that varies in length of 3’UTR depending on the cancer cell. Our PCR data of TNS4 was different from the length described in the TargetScanHuman database (shorter than the database). There may be no target sequences controlled by the miR-150-5p (guide strand). Detailed analysis is required for each cell. However, this analysis data is omitted this time.

Comment-5: Regarding Figure 4: the authors demonstrate the regulation of TNS4 by miR-150-3p well, but do not acknowledge the fact that the miR-150-3p binding site is at the end of the around 4600 nt long TNS4 3’UTR. Tumors are known for sometimes shortening some mRNAs 3’UTRs to secure them of posttranscriptional control. Did the authors check for the TNS4-3’UTR lengths in LUAD, for instance by qPCRs using differently located primers on the 3’UTR?

Response: Reviewer’s comment is an important issue for this paper. I checked the length of 3’UTR of TNS4 by several primer sets using oligo(dT) priming cDNA in A549 (LUAD) cells. Our data was different from the TargetScanHuman database, with the 3’UTR length of approximately 2000 bp. Target sequence of miR-150-3p has existed at the 3’end of the UTR. It has been confirmed that the target sequence of miR-150-3p is present in the 3'UTR of TNS4. However, it is necessary to clarify the sequence of 3'UTR of TNS4. This analysis is omitted this time. Along with this, Figure 4 and 2.5 sentences have been changed.

Binding site for miR-150-3p in the 3ʼ-UTR of TNS4 (positions 4473-4480 bp, 5’-UGUACCA-3’) were predicted by the TargetScanHuman database. Our data showed that several variants of the 3'UTR of TNS4 existed in cancer cells (data not shown). In A549 cell, it was confirmed that major variant of 3'UTR approximately 2 kb in length were significant, and a putative binding site of miR-150-3p existed in this variant (data not shown). Detailed analysis of TNS4 variants is necessary in each cancer cell.

Comment-6: Regarding chapter 2.6: were the LUAD specimens analyzed by immunohistochemistry corresponding to that analyzed for miR-150 expression in chapter 2.1.? Because in this chapter, 20 LUAD specimens were analyzed, compared to 18 in chapter 2.1. Please clarify the difference.

Response: I thank you very much for pointing that out.

The clinical specimens used for each experiment (2.1: miRNA expression and 2.6: TNS4 immunohistochemistry) are different. Expression data (miR-150-5p and miR-150-3p) are clinical samples collected at Kagoshima University Hospitals. On the other hand, immunostaining of TNS4 is analysis using a tissue microarray. Clinical information on the tissue microarray (Figure-5) is shown in the Supplemental table-2. The following text has been added to avoid confusion.

2.6. Expression of TNS4 protein in clinical LUAD specimens

Analysis using a tissue microarray was performed to examine TNS4 expression at the protein level. We validated the expression of TNS4 by using immunohistochemical staining. In this study, we stained 20 LUAD specimens and 14 non-cancerous specimens. Clinical information on the tissue microarray (Figure 5) is shown in the Table S2. Compared with non-cancerous tissues, TNS4 proteins were highly expressed in LUAD specimens (Figure 5).

Comment-7: There are several sentences scattered over the text, which are rather unspecific. For instance: l.26-28, l.201-203, l.250-252. I suggest to sharpen such rather broad, sometimes – I apologize - exaggerating statements.

Response: According to the reviewer’s comment, I rewrote the pointed out sentence.

Our approach, discovery of anti-tumor miRNAs and their target RNAs for LUAD, will contribute to the elucidation of molecular networks involved in the malignant transformation of LUAD.

The involvement of passenger strands of miRNAs in cancer pathogenesis is an attractive proposal for cancer research. Identification of novel molecules controlled by miRNA (the passenger strand of miRNA duplex) will contribute to the understanding of the oncogenic networks of LUAD.

The elucidation of the novel targets controlled by anti-tumor miRNAs will accelerate comprehensive understanding of oncogenic networks of LUAD.

Comment-8: Minor issue: in the introduction, the term “contributed to” is sometimes used false (l.47, l.50, etc.) – please correct.

Response: We have corrected the grammatical issues and misspelling pointed out in the manuscript.

Comment-9: Minor issue: typo in l. 112: transfected.

Response: We have corrected the grammatical issues and misspelling pointed out in the manuscript.

Comment-10: Minor issue: the Material and Methods section is quite short. Maybe the authors may give some more details to the reader rather than only redirect to their previous publications.

Response: Thank you for your comments. At first, the material and methods were described in detail. However, we were removed from the manuscript because it was pointed out that there were many overlaps.

Our data are the first to report these findings in LUAD. We believe that this article contributes significantly to our understanding of the molecular mechanisms of LUAD pathogenesis. We appreciate your consideration of this manuscript for publication in Cancers, Special Issue: "Molecular Profiling of Lung Cancer"

Sincerely,

Naohiko Seki, PhD

Department of Functional Genomics

Graduate School of Medicine, Chiba University

1-8-1 Inohana, Chuo-ku, Chiba 260-8670, Japan

Email: naoseki@faculty.chiba-u.jp

Reviewer 2 Report

Misono et al. present an interesting study concerning the mechanisms of gene regulation by miR-150 in LUAD. The Authors show that miR-150-3p downregulates TNS4 and implicate this gene in the aggressive behaviour of that type of cancer. Other works have documented the tumour suppressive role of miR-150, namely in LUAD. However, the studies focused on the passenger miR-150-3p are indeed novel and the relation between miR-150 and TNS4 is also a new and important finding. The work is well-structured, employs adequate methodologies and shows convincing results. The standards of English language need improvement.

Overall, the article contains significant novelty, and should be of interest for the readers of the Cancers. A few comments are listed below.

page 2, lines 46-48

Parts of the sentence are unclear. I suggest rephrasing to "...expression of miRNAs closely contributes to the pathogenesis of human diseases...".

lines 68-69

I suggest rephrasing to "...identify their targets with close associations with LUAD tumorigenesis."

lines 71-71

I suggest rephrasing to "...significantly attenuated the malignant phenotypes of cancer cells."

lines 73-74

Please delete the sentence: "Our ongoing studies, anti-tumor miRNAs-mediated ... malignant disease."

line 76

I suggest rephrasing to: "...miR-150-3p are downregulated..."

Figure 1- Please show the number of patients at risk for each time point (12, 24, 36 etc) in figures D and E. This can be shown below the X axis.

page 3, line 96

I suggest rephrasing to: "...miR-150-3p inhibits cancer..."

page 4, line 115:

I suggest rephrasing to: "Candidate target genes of miR-150-5p and miR-150-3p in LUAD: clinical significance of TNS4, SFXN1, SKA3 and SPOCK1 expression".

page 5, line 117

I suggest rephrasing to: "...was shown..." to "...is shown...".

lines 118-119

I suggest rephrasing to: "...oncogenic targets regulated by miR-150-5p and miR-150-3p were identified in LUAD cells..."

Tables 2 and 3 - first column should read Entrez, not Enterz

page 7, lines 125-126

This kind of survival analysis is not really about pathogenesis (the genesis of the disease) but more about disease outcomes. I suggest rephrasing: "... that were strongly associated with patient outcome (5-year overall survival:..."

page 8, lines 154-155

Please rephrase: "...non-cancerous tissues, TNS4 was highly expressed in LUAD specimens."

page 9, line 162

"suppresses" instead of "suppressed"

page 14, lines 195-203

I suggest rephrasing the whole paragraph: "According to the current concept of miRNA biogenesis, miRNA passenger strands are degraded and have no cellular functions. In contrast to this concept, our miRNA signatures based on RNA-sequencing revealed that some passenger strands are aberrantly expressed in cancer tissues [10, 16, 21, 24-27]. Importantly, functional assays (...) anti-tumor miRNAs by controlling cancer-related genes [17, 25, 28-33]. The involvement of miRNA passenger strands is an attractive proposal for cancer research and is likely to contribute for clarifying the molecular pathogenesis of LUAD."

lines 208-211

I suggest rephrasing to "Moreover, we revealed that miR-150-3p targets oncogenes involved in focal adhesions (...) their high expression was significantly correlated with poor prognosis [21-23].

lines 212-213

The notion of "oncogenic genes" appears multiple times in the manuscript. I believe we can conveniently refer to those simply as "oncogenes".

line 213:

rephrase to: "...SPOCK1) regulated by miR-150-3p in LUAD cells."

line 214:

Although the following sentences refer to previous work showing the involvement of these genes with a malignant phenotype, in this particular work, the Authors seem to have associated their four oncogenes with patient outcome (survival), rather than with pathogenesis. Please make sure this is made clear throughout the manuscript.

line 238: "... knockdown of TNS4. A total..."

lines 248-251: I suggest: "Thus, the data (...) by TNS4 are closely involved in the pathogenesis of cancer. The elucidation of novel RNA networks controlled by anti-tumor miRNAs will accelerate the comprehensive understanding of the molecular pathogenesis of LUAD."

page 20, lines 274-275

Why was miR-26a chosen for normalization?

Author Response

Cancers-467826

Cancers: Special Issue "Molecular Profiling of Lung Cancer"

Revised letter

April 10, 2019

Dr. Kentaro Inamura

Guest Editor

Ms. Diana Wang

Assistant Editor

Dear Dr. Inamura:

We would like to express our gratitude for your consideration of our above-mentioned manuscript for publication in Cancers. Enclosed, please find the revised manuscript Cancers (Cancers-467826) along with a detailed explanation of the revisions, which were made based on the reviewers’ comments. All changes are highlighted in the revised manuscript.

Reviewer #2

Comment: Figure 1- Please show the number of patients at risk for each time point (12, 24, 36 etc) in figures D and E. This can be shown below the X axis.

Response: As suggested by the reviewer’s comment, I added the number of patients at risk for each time point and modified the Figure-1.

Comment: page 20, lines 274-275, Why was miR-26a chosen for normalization?

Response: MiR-26a was chosen as an endogenous control because it is constitutively expressed in lung cancer tissues and does not affect miR-150-5p and miR-150-3p. Similar assays were performed using miR-21-5p and miR-16-5p, as endogenous controls, which are constitutively expressed in lung cancer tissues. The same results as miR-26a were obtained (Supplemental Figure 2).

In addition, the suggested key terms have been used throughout the manuscript and accompanying figures and tables.

Our data are the first to report these findings in LUAD. We believe that this article contributes significantly to our understanding of the molecular mechanisms of LUAD pathogenesis. We appreciate your consideration of this manuscript for publication in Cancers, Special Issue: "Molecular Profiling of Lung Cancer"

Sincerely,

Naohiko Seki, PhD

Department of Functional Genomics

Graduate School of Medicine, Chiba University

1-8-1 Inohana, Chuo-ku, Chiba 260-8670, Japan

Email: naoseki@faculty.chiba-u.jp

Reviewer 3 Report

The study evaluated the biological role of miR-150 (guide and passenger strands) in the pathogenesis of lung adenocarcinoma (LUAD). The level of both miR-150-5p and miR-150-3p was evaluated in LUAD tissues and cancer cells and the prognostic value also evaluated. Next gain-of-function showed the biological role of miR-150 in suppressing cell proliferation and invasion. MiR-150 transfected A549 cells were immuno-precipitated with Ago2 and target genes identified. Among them, tensin 4 (TNS4) was a direct target of miR-150 and a prognostic factor for LUAD. Overexpression of TNS4 was observed in cancer specimens and its silencing induced inhibition of cell growth and invasion, which were reversed by TNS4 expression vector.

The study is well performed and well presented. The methods used are appropriated and support the role of miR-150 and its direct target TNS4 in the pathogenesis of LUAD.   

Author Response

Cancers-467826

Cancers: Special Issue "Molecular Profiling of Lung Cancer"

Revised letter

April 10, 2019

Dr. Kentaro Inamura

Guest Editor

Ms. Diana Wang

Assistant Editor

Dear Dr. Inamura:

We would like to express our gratitude for your consideration of our above-mentioned manuscript for publication in Cancers. Enclosed, please find the revised manuscript Cancers (Cancers-467826) along with a detailed explanation of the revisions, which were made based on the reviewers’ comments. All changes are highlighted in the revised manuscript.

Reviewer #3

The study evaluated the biological role of miR-150 (guide and passenger strands) in the pathogenesis of lung adenocarcinoma (LUAD). The level of both miR-150-5p and miR-150-3p was evaluated in LUAD tissues and cancer cells and the prognostic value also evaluated. Next gain-of-function showed the biological role of miR-150 in suppressing cell proliferation and invasion. MiR-150 transfected A549 cells were immuno-precipitated with Ago2 and target genes identified. Among them, tensin 4 (TNS4) was a direct target of miR-150 and a prognostic factor for LUAD. Overexpression of TNS4 was observed in cancer specimens and its silencing induced inhibition of cell growth and invasion, which were reversed by TNS4 expression vector.

The study is well performed and well presented. The methods used are appropriated and support the role of miR-150 and its direct target TNS4 in the pathogenesis of LUAD.   

Response: We thank referees for careful reading our manuscript and for giving useful comments.

Our data are the first to report these findings in LUAD. We believe that this article contributes significantly to our understanding of the molecular mechanisms of LUAD pathogenesis. We appreciate your consideration of this manuscript for publication in Cancers, Special Issue: "Molecular Profiling of Lung Cancer"

Sincerely,

Naohiko Seki, PhD

Department of Functional Genomics

Graduate School of Medicine, Chiba University

1-8-1 Inohana, Chuo-ku, Chiba 260-8670, Japan

Email: naoseki@faculty.chiba-u.jp

Round 2

Reviewer 1 Report

The manuscript entitled „Molecular pathogenesis of gene regulation by the miR-150 duplex: miR-150-3p regulates TNS4 in lung adenocarcinoma“ by Misono and colleagues covers the role of miR-150-3p in the disease progression of lung adenocarcinoma. The author describe the passenger strand of the miR-150 duplex to be downregulated in the analyzed LUAD patients samples in comparison to non-cancerous tissues, analogous to the guide strand miR-150-5p. Furthermore, the authors identify several putatively miR-150-3p regulated target genes associated with LUAD progression and confirm tensin 4 (TNS4) as miR-150-3p target gene by luciferase reporter assay and western blot. Finally, the authors demonstrate TNS4 overexpression to enhance the LUAD aggressiveness, and identify 88 potentially TNS4-regulated genes, many of them associated to LUAD disease progression. This is the first revision of the manuscript. I appreciate the detailed answers to my remarks, as well as the additional experiments conducted by the authors. However, I have some additional comments on the answers to my comments, in the following point by point:

Comments 1,3,7-9,10: Thank you for the corrections of clarifications, I do not have any remarks to these points.

Comment 2: I appreciate the additional “dilution” experiments conducted by the authors, showing that the 1 nM of miR-155-3p exhibit similar significant effects on proliferation, migration or invasion as 10 nM miR-155-5p or -3p. However, the fact remains that according to the authors’ experiments miR-155-3p is 100-1000x lower expressed in LUAD than miR-155-5p (similar numbers in mirbase.org: miR-155-5p: miR-155-3p = 1:0.05, comparing the NGS reads). To clarify my initial comment: the authors should mention this fact in the results section, and maybe discuss the biological implications in a separate section in the discussion, because – setting aside that in vitro assays have some limitations on their convertibility to in vivo statements – the fact remain that the results shown in S1 exhibit no significant tumorbiological effect of the 100x fold lower miR-155-3p concentration in comparison to the (arbitrarily set) 100% miR-155-5p concentration (10 nM).

Comment 4/5: I appreciate the additional experiments the authors conducted, and the sound explanation. However, the question for the regulation of TNS4 by miR-155-5p and the respective putative binding site in the 3’UTR is crucial (as 4 miR-155-5p binding sites >> 1 miR-155-3p binding site). To my opinion, the statement “There may be no target sequences controlled by the miR-150-5p (guide strand).” is – sorry – far too sloppy. Omitting the analysis data “this time” seems – even sorrier – suspicious. If the authors analyzed the TNS4 3’UTR by PCR, I see no obstacle in not analyzing the amplicon by sequencing. The retrieved sequence for A549 (and maybe H1299) could be table S3. Then, it would be possible to search for binding sequences (at least the seed sequence) of miR-155-5p in the retrieved sequence data, and at least get a glimpse where in the TNS4 sequence the 3’UTR-shortening happened (to clarify: only in the LUAD cell lines, which are applicable to the manuscript, not the other cell lines, which are nice to know, but a different story). And then – only then – I see the statement “there are no miR-150-5p target sites in the TNS4-3’UTR in our cell line” as justified (because in Homo sapiens, there is already less strong evidence gained by NGS, that miR-150-5p also controls TNS4 – see http://mirtarbase.mbc.nctu.edu.tw/php/detail.php?mirtid=MIRT652285#target ).     

Comment 6: I understand the origin of the analyzed specimen now. Of course, It would be much more interesting to have comparative data on TNS4 IHC and miR-155-3p (and maybe miR-155-5p) RNA in situ hybridisation in these specimen, but I understand that these experiments are too time- and resources-consuming to be conducted within this manuscript. Therefore, I thank the authors for the clarification within the manuscript.

Finally, I apologize for insisting of additional analyses(not necessarily new experiments) as crucial for this work; however, I am not fully convinced yet. Best regards.

Author Response

Cancers-467826

Cancers: Special Issue "Molecular Profiling of Lung Cancer"

Revised letter

April 23, 2019

Dr. Kentaro Inamura

Guest Editor

Ms. Diana Wang

Assistant Editor

Dear Dr. Inamura:

We would like to express our gratitude for your consideration of our above-mentioned manuscript for publication in Cancers. Enclosed, please find the re-revised manuscript Cancers (Cancers-467826) along with a detailed explanation of the revisions, which were made based on the reviewers’ comments. All changes are highlighted in the revised manuscript.

Comment 2: I appreciate the additional “dilution” experiments conducted by the authors, showing that the 1 nM of miR-155-3p exhibit similar significant effects on proliferation, migration or invasion as 10 nM miR-155-5p or -3p. However, the fact remains that according to the authors’ experiments miR-155-3p is 100-1000x lower expressed in LUAD than miR-155-5p (similar numbers in mirbase.org: miR-155-5p: miR-155-3p = 1:0.05, comparing the NGS reads). To clarify my initial comment: the authors should mention this fact in the results section, and maybe discuss the biological implications in a separate section in the discussion, because – setting aside that in vitro assays have some limitations on their convertibility to in vivo statements – the fact remain that the results shown in S1 exhibit no significant tumorbiological effect of the 100x fold lower miR-155-3p concentration in comparison to the (arbitrarily set) 100% miR-155-5p concentration (10 nM).

Response: As suggested by the reviewer’s comment, I modified sentences (yellow highlights in the text) and added following sentences in the Discussion as follows.

In general theory, passenger strand of miRNAs derived from miRNA-duplex have degraded in cytoplasmic and have no function. In fact, the expression of passenger strand of miRNA is overwhelmingly lower than that of guide strand. Are passenger strand of miRNA really functional in vivo? It is an important issue in miRNA research. Expression levels of miR-150-3p was lower (100X) than miR-150-5p in LUAD cells. Our in vitro functional assays showed that antitumor effects were observed even if the transfection concentration of mature miR-150-3p is lowered (1 nM and 0.1 nM). Elucidation of functions of passenger strand of miRNA in vivo is an important biological theme. The involvement of passenger strands of miRNAs in cancer pathogenesis is an attractive proposal for cancer research.Identification of novel molecules controlled by miRNA (the passenger strand of miRNA duplex) will contribute to the understanding of the oncogenic networks of LUAD. Involvement of passenger strands of miRNAs in cancer pathogenesis is an attractive proposal for cancer research. Identification of miRNAs (passenger strands of miRNA-duplex) will contribute to the understanding of new molecular pathogenesis of LUAD.

Comment 4/5: I appreciate the additional experiments the authors conducted, and the sound explanation. However, the question for the regulation of TNS4 by miR-155-5p and the respective putative binding site in the 3’UTR is crucial (as 4 miR-155-5p binding sites >> 1 miR-155-3p binding site). To my opinion, the statement “There may be no target sequences controlled by the miR-150-5p (guide strand).” is – sorry – far too sloppy. Omitting the analysis data “this time” seems – even sorrier – suspicious. If the authors analyzed the TNS4 3’UTR by PCR, I see no obstacle in not analyzing the amplicon by sequencing. The retrieved sequence for A549 (and maybe H1299) could be table S3. Then, it would be possible to search for binding sequences (at least the seed sequence) of miR-155-5p in the retrieved sequence data, and at least get a glimpse where in the TNS4 sequence the 3’UTR-shortening happened (to clarify: only in the LUAD cell lines, which are applicable to the manuscript, not the other cell lines, which are nice to know, but a different story). And then – only then – I see the statement “there are no miR-150-5p target sites in the TNS4-3’UTR in our cell line” as justified (because in Homo sapiens, there is already less strong evidence gained by NGS, that miR-150-5p also controls TNS4 – see http://mirtarbase.mbc.nctu.edu.tw/php/detail.php?mirtid=MIRT652285#target).   

Response: In order to clear this problem (the presence or absence of the target sites of miR-150-5p), I decided to clone the 3’UTR of TNS4 and confirm the sequences.

It has become clear that the 3’UTR of TNS4 has several variants. The nucleotide sequence was determined for the major variants. Add this sequence data of 3’UTR of TNS4. Although the binding site of miR-150-3p exists in the sequence, the putative binding sites of miR-150-5p do not exist.

It is an important issue to confirm the nucleotide sequences of 3'UTR about the cell line used for analysis.

The LUAD cell line (A549) used this study was different from the expected sequences of 3’UTR of TNS4 by TargetScanHuman database.

I re-analyzed luciferase reporter assay to investigate direct binding of miR-150-3p in 3’UTR of TNS4 gene based on our sequencing data.

Our data showed that miR-150-3p was directly binding miR-150-3p in 3’UTR of TNS4 in A549 cells. This data is incorporated into Figure 4, and modified points are shown in yellow highlights in the text.

In order to confirm the binding site of miR-150-3p, the nucleotide sequences of 3’UTR of TNS4 in A549 cells was examined independently. Our data showed thatseveral variants of the 3'UTR ofTNS4existed inA549 cells (Figures S5). As a result of sequencing analyses, one putative binding site of the miR-150-3p was found in 3’UTR of TNS4 (Figures S5). Based on our sequence data, we used luciferase reporter assays with vectors carrying either the wild-type or deletion-type 3’-UTR of TNS4 (Figure 4C). We observed significantly reduced luminescence intensities after transfection with miR-150-3p and the wild-type 3’-UTR of TNS4 (Figure 4C). Transfection with the deletion-type vector did not reduced luminescence intensities in A549 cells (Figure 4C). Thus, miR-150-3p directly bound to TNS4 in the 3’-UTR. Although, TargetScanHuman database predicted putative binding sites of miR-150-5p in 3’UTR of TNS4, our sequencing data could not confirm the sequences.

Comment 6: I understand the origin of the analyzed specimen now. Of course, It would be much more interesting to have comparative data on TNS4 IHC and miR-155-3p (and maybe miR-155-5p) RNA in situ hybridization in these specimens, but I understand that these experiments are too time- and resources-consuming to be conducted within this manuscript. Therefore, I thank the authors for the clarification within the manuscript.

Response: I appreciate for your helpful comment for our study.

We will add researcher (Shogo Moriya, PhD) who have made significant contributions to the previous and current revised experiments to the author of this article. All authors of this paper agree on this matter.

We believe that this article contributes significantly to our understanding of the molecular mechanisms of LUAD pathogenesis. We appreciate your consideration of this manuscript for publication in Cancers, Special Issue: "Molecular Profiling of Lung Cancer”.

Sincerely,

Naohiko Seki, PhD

Department of Functional Genomics

Graduate School of Medicine, Chiba University

1-8-1 Inohana, Chuo-ku, Chiba 260-8670, Japan

Email: naoseki@faculty.chiba-u.jp

Round 3

Reviewer 1 Report

I want to thank the authors for the additional experiments they performed and the clarifications in the manuscript. I have no further requests; however some additional comments:

Regarding the TNS4 3'UTR: thank you for the inclusion of the alternative 3'UTRs determined by sequencing in the LUAD cell lines. Interestingly, these 3'UTRs differ from the NCBI sequence (NM_032865.6) especially in their 3'-regions (according to a BLAST alignment: NM_032865.6 - Variant 1: 12% identity, variant 2: 45% identity, variant 3: 46% identity). Although all three variants showed 100% similarity to chromosome 17 variants, there seems to be different splicing/transcription processes leading to these different forms. It would be interesting - for further experiments - to analyze the underlying mechanisms, given the fact, that the proposed mir-150-3p binding site does not seem to be part of the "official" human NCBI TNS4 cds sequence (NM_032865.6).

I want to thank the authors for sharing your work with the scientific community.

Best regards.